# Detection of Psychopathic Traits in Emotional Faces

**DOI:** 10.3390/jintelligence9020029

**Published:** 2021-06-04

**Authors:** Sally Olderbak, Christina Bader, Nicole Hauser, Sabina Kleitman

**Affiliations:** 1Institute of Psychology and Education, Ulm University, Albert-Einstein-Allee 47, 89081 Ulm, Germany; 2Therapy Department, Justizvollzugsanstalt Bruchsal, Schönbornstraße 32, 76646 Bruchsal, Germany; christina.bader@gmx.de; 3Department of Forensic Psychiatry, University Hospital of Zurich, Rämistrasse 100, 8006 Zürich, Switzerland; nicole.hauser@puk.zh.ch; 4School of Psychology, University of Sydney, Camperdown, NSW 2050, Australia; sabina.kleitman@sydney.edu.au

**Keywords:** psychopathy, perception, halo effect, detecting psychopathy, thin-slice judgments

## Abstract

When meeting someone at zero acquaintance, we make assumptions about each other that encompass emotional states, personality traits, and even cognitive abilities. Evidence suggests individuals can accurately detect psychopathic personality traits in strangers based on short video clips or photographs of faces. We present an in-depth examination of this ability. In two studies, we investigated whether high psychopathy traits are perceivable and whether other traits affect ratings of psychopathic traits in the sense of a halo effect. On the perceiver’s end, we additionally examined how cognitive abilities and personality traits of the responders affect these ratings. In two studies (*n*_1_ = 170 community adults from the USA, *n*_2_ = 126 students from Australia), participants rated several targets on several characteristics of psychopathy, as well as on attractiveness, masculinity, sympathy, trustworthiness, neuroticism, intelligence, and extraversion. Results show that responders were generally able to detect psychopathy. Responders generally came to a consensus in their ratings, and using profile similarity metrics, we found a weak relation between ratings of psychopathy and the targets’ psychopathy level as determined by the Psychopathy Checklist: Short Version. Trait ratings, though, were influenced by the ratings of other traits like attractiveness. Finally, we found accuracy in the perception of psychopathy was positively related to fluid intelligence but unrelated to emotion perception ability.

## 1. Introduction

The first impression colloquially counts, which is supported by research reporting that a few seconds of exposure to a stranger are enough to judge many traits ([1]). For example, trustworthiness ([53]), extraversion ([26]), and intelligence ([39]; [55]; [44]) are routinely found to be perceivable based on a brief observation. From an evolutionary perspective, the quick forming of an opinion about a social interaction partner is seen as essential ([19]; [2]) and dates back to Darwin, who attributed the perception of emotions in others as important for survival ([9]). The perception of traits in others at zero acquaintance, known as a thin slice, is an interpersonal process that can be dissected from different perspectives (e.g., see Cronbach’s componential approach ([8]) or Funder’s Realistic Accuracy Model ([15])).

The desire to know about those with whom we interact is especially relevant for the perception of personality traits associated with a disregard for the welfare of others, as is typical for psychopathic personality traits ([16]). Highly psychopathic individuals present interpersonal/affective disturbances (e.g., callous and manipulative traits) with an antisocial lifestyle (e.g., poor behavioral control and impulsivity) that often results in law-breaking behavior as well as in hurting others during social interactions ([16]; [17]). In addition to committing a disproportionate number of violent crimes, these individuals are also responsible for the use of instrumental aggression towards others to achieve their goals ([54]). Thus, the capacity to detect a highly psychopathic person would be important for survival.

However, the prevalence of high psychopathic traits in the general population is only about 1% ([6]; [22]). It is argued that at a higher prevalence, highly psychopathic persons would be more readily detected, and thus their parasitic lifestyle would be ineffective ([25]). Thus, it is important for highly psychopathic persons that they are not perceived as such. Nevertheless, several studies have found that psychopathic traits can be detected based on thin slices (e.g., [23]; [14]; [46]; [35]; [3]). 

## 2. Perception of Psychopathy

With his study on the detection of dark triad traits from faces, [23] ([23]) found that people were able to successfully detect psychopathic traits in neutral human faces. The targets he used were students whose pictures were digitally merged to create prototypes of men and women who self-reported being high or low on psychopathy and the other dark triad traits. A second study showed similar results using short videos of prison inmates who were participating in the Psychopathy Checklist Interview-Revised ([14]). In this study, it was shown that raters performed better with less information about the target, supporting the idea that detecting psychopathy based on a 5-second video clip about the target, even a first glance only, is possible. Finally, a third study showed participants were accurate in their perception of subclinical psychopathy when given a 1.5-minute video with audio ([35]). 

This research is important because it suggests that personality traits linked to potential violence are perceivable at zero acquaintance. Indeed, when presented with a choice, women preferred facial morphs based on men low in psychopathy, particularly those they perceived to be not dangerous. They were rated as more attractive relative to facial morphs based on men high in psychopathy ([46]; [3]). 

However, a closer look at the study designs suggests several questions remain. [23] ([23]) used target faces of a student sample with low between-subject variance on psychopathy and because it was a student sample, generally lower levels of psychopathy. Most studies rely on neutral expressions in target faces, neglecting the emotional information value of interpersonal encounters (e.g., [23]). 

Some used video clips of forensic target subjects (e.g., [14] or [35]). However, the use of video clips is possibly a source of confounding variables. In particular, the manipulative and superficially charming interaction style of highly psychopathic individuals is given a chance to be activated in videos involving verbal behavior and mimics that could already influence ratings of observers. These parameters complicate drawing conclusions about the detection of personality traits at zero acquaintance. A more recent study took these findings to explore the influences on those impressions and found that higher psychopathy scores were associated with a distinctive communication style ([47]). Especially the smiles of the targets, negative emotional language, and hand gestures influenced the observer’s impressions.

These studies also neglect the possibility that other variables, such as the attractiveness or perceived masculinity of the targets, impacted ratings. Referred to as halo effects, these should always be considered when rating targets based on appearance (e.g., less attractive persons may be rated as more psychopathic). 

Additionally, the method upon which accuracy estimates were based is important. [14] ([14]) compared rating against the targets’ psychopathy ratings as determined through the Psychopathy Checklist-Revised (PCL-R ([16])), an established assessment tool that is scored based on a review of criminal records and an interview by a trained interviewer. However, [23] ([23]) and [35] ([35]) assessed psychopathy levels through self-reported responses of the targets, which are influenced by social desirability and other biases. Finally, to our knowledge, no one has yet investigated agreements in responders’ ratings and the extent to which ratings could be attributable to whom was being rated. 

## 3. Current Study

Thus, while some have concluded that psychopathy can be perceived at zero acquaintance, we argue that there are still several questions to be addressed. In this study, we present a critical examination of this topic and investigate alternative explanations for the findings by [23] ([23]) and others. 

### 3.1. Research Question 1: Can Psychopathy Be Perceived?

First, we revisit the basic question of whether or not psychopathy can be perceived. We will investigate whether there is a significant and stable portion of variance in the perception of psychopathy that can be attributed to the target. A significant portion indicates agreement between the responders on their ratings of the targets and that some targets are consistently rated highly across all responders while other targets are consistently rated lower. Thus, a significant proportion of variance attributable to targets would mean that there is an aspect of the target that leads responders to reliably rate the targets in a specific way.

This shared agreement, however, could indicate a shared bias by the perceivers. Thus, we will then additionally compare the ratings against a veridical indicator of the targets’ psychopathy. There are several options for the choice of a veridical response. First, one could use scores self-reported by the target; however, they may be unaware of their trait levels or motivated to lie. Second, several outside observers could provide a consensus rating; however, they may be insufficiently informed, or have poor trait perception abilities themselves. Instead, we sought a third option, namely obtaining an objective external assessment through the use of a standardized test. Specifically, we used ratings from the Psychopathy Checklist: Short Version (PCL:SV ([18])) as a veridical indicator of the targets’ psychopathy. We then estimated the general agreement between the ratings. 

### 3.2. Research Question 2: Is the Perception of Psychopathy Due to Halo Effects?

During thin slice judgments, individuals’ perceptions of one trait will impact how other traits are perceived. This is referred to as the halo effect ([48]). Namely, the halo of one positively perceived attribute such as attractiveness biases the further evaluation of other objectively independent characteristics such as intelligence so that individuals perceived to be attractive are also perceived to be higher in intelligence ([30]; [13]; [10]). A halo effect is also observable for the perception of emotions, such that a smiling person is perceived as a morally good and intelligent person ([31]).

Given the prevalence of halo effects, it is important to investigate whether this also impacts the perception of psychopathy. For example, are persons who are less attractive perceived to be higher on psychopathy? Is the perception of intelligence impacting perceptions of psychopathy? We will specifically investigate the impact of perceptions of attractiveness, masculinity, intelligence, trustworthiness, extraversion, sympathy, and responsibility as causing halo effects in the perception of psychopathy. 

### 3.3. Research Question 3: Is Psychopathy Perception a Cognitive Ability?

Assuming that psychopathy can be perceived and that ratings are not due to halo effects, we will then investigate whether there are individual differences in the accuracy by which one perceives psychopathy. Specifically, we will test whether accuracy in perception is correlated with other cognitive abilities such as general fluid intelligence and emotion perception ability ([36]). Assuming psychopathy perception is a cognitive ability, we hypothesize that accuracy in perception will be positively correlated with both abilities but more strongly related to emotion perception ability because it is also considered a social intelligence. 

Thus, in two studies, we advance the present literature by investigating the perception of thin slice facial emotional expressions from persons from correctional facilities and a community sample that were previously rated on the psychopathy continuum. We evaluate agreement between raters, the proportion of variance in ratings attributable to the target, the influence of halo effects, and whether accuracy is also related to cognitive abilities.

## 4. Study 1

### 4.1. Methods

#### 4.1.1. Perceiver Sample

A total of 331 participants started the study. We removed *n* = 54 persons, because they had no variance in their ratings across all faces, and an additional *n* = 107, because they rated less than one-third of the targets, resulting in *n* = 170 participants. Of this sample, *n* = 89 rated all 30 targets. We checked whether there were differences in the within-target standard deviations, mean psychopathy ratings, or mean trait ratings (i.e., all characteristics) for the first 10 targets between those who did and did not rate all 30 faces. There were no significant differences between the groups. Thus, we allowed participants with missing data, including all participants who rated at least 10 faces, leaving us with a final sample of *n* = 170. However, please note that the results presented below are comparable when estimated with the *n* = 89 sample. With these sample sizes, we have the power (two-tailed *p* value, α = 0.05, 1- β = 0.80) to detect small (*r* = 0.21, *n* = 170) to medium effects (*r* = 0.21, *n* = 89).

#### 4.1.2. Procedure

The study was presented online using Unipark from Questback. Participants were recruited from the United States of America population via MTurk. Mturk is an efficient tool to obtain data that stem from a diverse sample, and the data are comparable to samples collected with traditional methods ([4]; [5]). Participants individually provided informed consent and data was collected in accordance with the hosting University’s regulations and the Declaration of Helsinki. All tasks were presented in the same order for each participant. 

#### 4.1.3. Target Creation

The targets were selected from a sample of men (age 18–60 years) who participated in a larger study investigating socio-emotional traits and psychopathy (n = 339; see [40]; [28]; and [42] for more details about this sample). The target sample consists of prison inmates, nurses, and a community sample who completed an emotion expression task during which they were asked to make a neutral face or pose an expression of one of the six basic emotions (anger, disgust, fear, happiness, sadness, surprise) ([11]). During the task, the men’s faces were videotaped, and their expressions were rated using an emotion expression coding software, [12] ([12]). For study 1, 30 men were selected who gave consent to use their photographs for our task. We also selected those whose faces were well lit and had sufficient resolution. The mean age of the targets was *M* = 36.75 (*SD* = 9.5), 93.3% of them identified as German, and all targets were Caucasian. Their education ranged from high school to university degree, with most of them (66.7%) completing high school. For this task, we selected images where 10 men expressed a neutral face, 10 expressed anger, and 10 expressed happiness. That these emotions were expressed was confirmed through their corresponding Emotient scores.

Targets also completed the Psychopathy Checklist: Short Version (PCL:SV ([18])) for an assessment of their psychopathy levels. The PCL:SV consists of 12 items for which an interviewer assigns a score of 0 (no), 1 (to some extent), and 2 (yes). Ratings are made based on a semi-structured interview and available criminal records. The PCL:SV measures two superordinate factors of psychopathy: the core personality traits (Factor 1) and social deviance (Factor 2). We used a cut-off score of 15 or higher, in accordance with recommendations for German samples ([38]), to identify a person as highly psychopathic. Based on the ratings of 13 cases, the inter-rater agreement of PCL:SV ratings was substantial ICC(1,1) = 0.80 [95% CI = 48, 0.93] ([29]). Likewise, another 21 participants had PCL-R scores available through their criminal records. These scores were multiplied by 3/5 so they would be analogous to the PCL:SV range. The inter-rater agreement at 0.75 (95% CI = [0.48, 0.89]) was also substantial. The target PCL:SV scores ranged from 0 to 22 (*M* = 11.17 *SD* = 6.83). 

The targets’ fluid intelligence was assessed with the “Berliner Test zur Erfassung fluider und kristalliner Intelligenz” or BEFKI Figural task ([50]). The 16-trial task measures fluid intelligence, and participants are asked which pattern completes a series. The task is scored as correct or incorrect and the measure had acceptable internal consistency (α = 0.68; ω_Total_ = 0.69). Average performance (*M* = 0.46, *SD* = 0.24) was sufficiently above guessing probability (0.11).

We received ethical approval for the target sample from the ethics committee of the Department of Psychology at Humboldt University of Berlin (application numbers 2012–40 and 2014–06), in addition to permission from the institutional review boards of the participating forensic psychiatric hospitals and prisons. Participants’ individual informed consent was attained through a series of consent forms.

#### 4.1.4. Measures

**Person Perception Task.** Using a half-block social relations design ([27]; [45]), every responder viewed every target (*n* = 30; see Figure 1) and rated each target on 16 traits. The traits included the 12 PCL:SV items in addition to *attractiveness, intelligence, masculinity*, and *trustworthiness*. The items *attractiveness* and *masculinity* were chosen because they give an indicator of the physical appearance of our targets. *Trustworthiness* was included because it can be seen as the opposite of psychopathy. We included the item *intelligence* because it has been shown in the literature that this item can be accurately perceived in strangers ([39]; [55]; [44]). Faces were displayed until participants rated all characteristics and selected to move on to the next item. The order of targets was the same for each perceiver. 

Participants were asked to rate each item along with a scale ranging from 1 (not at all) to 7 (absolutely). Prior to the task, participants were given definitions for each item. For example, the item *Impulsive* was defined as “acts without considering the consequences of his actions, acts on the spur of the moment out of excitement and the desire for risk”. Detailed instruction for this task and the definitions of the items are available upon request. We then computed an average across the PCL:SV items to create two psychopathy scores. We averaged responses to superficial, grandiose, deceitful, lacks remorse, lacks empathy, and does not accept responsibility to estimate ratings of the psychopathy superordinate factor, Factor 1, and impulsive, lacks goals, irresponsible, poor behavioral controls, adolescent antisocial behavior, and adult antisocial behavior to estimate ratings of the psychopathy superordinate factor, Factor 2. The two subscale scores had good internal consistency: Factor 1 (α = 0.96) and Factor 2 (α = 0.95). 

When creating this task, we designed the task so that there was variety between the faces in regard to which emotion was expressed, with one-third of the faces expressing happiness, one-third expressing anger, and one-third having a neutral expression. However, we did not have hypotheses about how the emotion expressed would impact trait ratings. Nevertheless, we conducted several exploratory analyses to examine the impact of emotion expressed on trait ratings. We found no strong difference, and because of the exploratory nature of those analyses, any significant differences were removed when controlling for multiple testing. Thus, the results below ignore emotion expressed as a moderator. 

**Identification of Emotion Expressions from Composite Faces (Emotion Composite).** With the Emotion Composite task, we measured the ability to perceive emotions in faces ([52]). Two photographs of a person expressing two different emotions are cut horizontally along the middle of the face and used to create a single face that is expressing two different emotions. Thus, the upper half of the face shows an expression of one emotion and the lower part of the face shows an expression of another emotion. Participants are then asked to identify the emotion expressed in the “Top” or “Bottom” half of the face. Expressions are limited to the six basic emotions (anger, disgust, fear, happiness, sadness, and surprise), and the task contains 72 trials. The task was scored with an unbiased hit rate ([49]) to control for personal biases in emotion perception, and using the UHR scores, it had strong internal consistency (α = 0.95).

### 4.2. Results

#### 4.2.1. Research Question 1: Can Psychopathy Be Perceived?

We applied social relations analysis ([36]; [45]) to estimate the proportion of variance attributable to the target and the perceiver. As mentioned earlier, significant variance attributable to the target indicates agreement amongst the perceivers, suggesting there is something perceived in the targets that causes them to be reliably perceived. Variance attributable to the perceiver effect indicates stable differences between perceivers in how they saw all targets. In other words, significant perceiver variance means some perceivers rated all targets as high on psychopathic traits whereas others rated all targets as low on psychopathic traits.

There was a significant proportion of variance attributable to the target for all six traits indicating there was something perceived in the target that predicted the responder’s ratings of the target (see Table 1). In addition to estimating whether the portion of variance was significantly different from zero, we also estimated the proportion of the total variance attributable to the target. Historically, a proportion of variance greater than 10% indicates that trait was perceived ([26]). The proportion of variance attributable to the target was generally low, and only psychopathy Factor 2 and attractiveness were above the 10% cut-off often used in the literature ([26]). 

A large proportion of variance for all traits was attributable to the responder, indicating significant individual differences in the perception of the targets, and that how responders perceived the targets could be largely attributable to the responders themselves.

Next, we examined the accuracy in trait ratings using profile similarity metrics. For each perceiver, we correlated their ratings of the target with the targets’ veridical scores for that variable based on their PCL:SV scores (Note: To ensure stable estimates, these analyses were conducted only for participants who rated all 30 faces, *n* = 89). Specifically, perceptions of psychopathy Factor 1 were correlated with the targets’ PCL:SV Factor 1 pro-rated scores, and perceptions of psychopathy Factor 2 were correlated with the targets’ PCL:SV Factor 2 pro-rated scores. 

On average, there was a positive correlation between perceivers’ ratings of the target and the targets’ psychopathy levels, as determined by their PCL:SV scores. However, the correlations were weak in magnitude and for individual persons, the correlations ranged widely from moderately negative to moderately positive (Psychopathy Factor 1: *M* = 0.12, ranged from −0.50 to 0.39; Psychopathy Factor 2: *M* = 0.10, ranged from −0.29 to 0.29). 

#### 4.2.2. Research Question 2: Is the Perception of Psychopathy Due to Halo Effects?

Next, we examined the extent to which ratings of psychopathy are related to ratings of other traits, referred to as halo effects. We investigated halo effects in the ratings of each psychopathy factor by regressing psychopathy ratings on ratings of traits, thus controlling for the impact of perception of those variables on the perception of psychopathy. We regressed psychopathy ratings on ratings of attractiveness, intelligence, trustworthiness, and masculinity in successive regressions. We then estimated a social relations analysis on the residuals of each regression (see Table 1). We found that for both psychopathy factors, the variance attributable to the target remained significantly different from zero. However, the percent of variance attributable to the target dropped to 5% for psychopathy Factor 1 and to 7% for psychopathy Factor 2. For both psychopathy factors, there was only a significant proportion of target variance when the targets were expressing happiness. Thus, there are halo effects in the perception of psychopathy. 

#### 4.2.3. Research Question 3: Is Psychopathy Perception a Cognitive Ability?

Furthermore, we examined whether accurately rating psychopathy was related to accurately rating other traits or with other cognitive abilities (Note: Again, to ensure stable estimates, these analyses were conducted only for participants who rated all 30 faces, *n* = 89). For these analyses, the perception of intelligence was correlated with the targets’ BEFKI figural task scores. On average, the correlation for the perception of intelligence was also weak in magnitude, with scores for individual persons ranging from moderately negative to moderately positive (*M* = 0.13, ranged from −0.44 to 0.46). 

Accuracy scores were correlated with each other and with emotion perception ability (see Table 2). Results show that accuracy in the perception of psychopathy Factor 1 was significantly correlated with accuracy in the perception of psychopathy Factor 2. Furthermore, accuracy in the perception of intelligence was significantly correlated with emotion perception ability. However, accuracy in the perception of either psychopathy factor was unrelated to accuracy in the perception of intelligence or emotion perception ability. 

### 4.3. Discussion Study 1

We could confirm the results of others, that psychopathy can be perceived. For both psychopathy factors, there was a significant proportion of the variance attributable to the target, indicating there was something about the faces that predicted similar ratings by the perceivers. Additionally, on average there was a weak positive correlation between participants’ ratings of the targets and the targets PCL:SV scores. However, there were strong individual differences in this ability, with numerous individuals giving high psychopathy ratings to men with low PCL:SV scores.

We also found evidence for halo effects in perceivers’ ratings of the targets, suggesting that psychopathic traits were not rated independently of other traits. Controlling for ratings of attractiveness, masculinity, and other traits lowered the proportion of variance attributable to the target for both psychopathy factors to below 10%.

Finally, we investigated whether trait perception is a cognitive ability. Overall, accuracy in trait perception ratings was positively related to emotion perception ability; however, the correlations were only statistically significant for accuracy in the perception of intelligence, and accuracy in the perception of psychopathy Factor 1 (assuming a one-tailed *p* value). Thus, we cannot conclude whether accuracy in trait perception is a cognitive ability. Average accuracy was quite low and most correlations with accuracy scores were not significant. However, in the present study assessment of cognition was limited to a single cognitive ability task, and because of the relatively small sample size (*n* = 89), we could not model relations at the latent level, which would have allowed us to control for unreliability. We addressed these limitations in the next study by increasing the number of participants, allowing for latent variable models, and by including an additional measure of the perceivers’ cognitive abilities. 

While we looked at psychopathy perception for Factors 1 and 2 separately, we did not find strong differences in the results for either factor. For both, there was a weak positive correlation between perceiver ratings and PCL:SV scores, and for both, there was a significant proportion of variance attributable to the target that was reduced to a comparable percentage when controlling for halo effects. Additionally, accuracy in the perception of psychopathy Factor 1 was significantly correlated with accuracy in the perception of psychopathy Factor 2, and both were weakly positively related to emotion perception ability. Thus, overall, we did not find noticeable differences in the extent to which Factor 1 and Factor 2 were perceived. It is not clear that one should expect separate abilities for the perception of each factor. Finally, there are alternative proposed structures to psychopathy, such as [7] ([7]) or Patrick and colleagues ([43]). Thus, for Study 2, we instead chose to look at the perception of psychopathy in general, rather than for specific psychopathy dimensions.

In study 1, we did not control for differences in the background images of the target or distractors on their face (e.g., presence of glasses). We addressed this limitation in study 2 by completely removing the background and choosing targets without accessories like glasses. Furthermore, because the emotion of the target did not impact results, we decided to not include emotion expressed as an aspect of the study design and instead presented faces with all emotions in the next study, to show the targets in different contexts. 

Finally, we found accuracy in the perception of psychopathy was unrelated to accuracy in the perception of intelligence, suggesting trait perception abilities may differ for individual traits. In study 2, we pursued this question further by looking at abilities for the perception of additional traits. 

## 5. Study 2

### 5.1. Methods

#### 5.1.1. Perceiver Sample

Participants were first year students, majoring in various faculties such as science, arts, business, and engineering, who were enrolled in a psychology course at the University of Sydney, Australia. The final sample consisted of *n* = 126 participants (73% females) who completed all relevant tasks for this study. The mean age was *M* = 19.86 (*SD* = 3.53), and the majority were born in Australia (66.7%) with English as their first language (77%). With this sample size, we have the power (two-tailed *p* value, α = 0.05, 1- β = 0.80) to detect small to medium effects (*r* = 0.25). 

#### 5.1.2. Procedure

The study was administered online and included several measures (see [32] for a full list). We will only discuss those relevant to our research question. Participants provided informed consent and ethics approval was obtained at the University of Sydney with Project Number: 2015/229. The order of tasks was counterbalanced across participants.

#### 5.1.3. Target Creation

The targets (*n* = 12) for this sample were chosen from the same participant pool as used in study 1. Given the responders were students from the University of Sydney, we controlled for age in the target group to fit with the age group of responders. The 12 targets selected had an age range of 18–30 (*M* = 24.35, *SD* = 3.79). We also selected targets whose PCL:SV scores were low (*n* = 4; PCL:SV sum score of 0 to 5), medium (*n* = 4; PCL:SV sum score of 6 to 13), and high (*n* = 4; PCL:SV sum score of 15–24). Of these men, 10 were German and 2 were non-German, and all of them were Caucasian. We excluded targets who were wearing glasses, who were wearing distracting accessories, or who were photographed in front of a context-specific background that made it obvious that they are in a prison of forensic psychiatry.

#### 5.1.4. Measures

**Person-Perception Task**. Whereas in study 1, only one photograph with one specific emotion was presented, in study 2, targets were presented expressing each of six basic emotions in addition to a neutral expression (see Figure 2). The final picture of the target showed all expressions next to each other, helping the responders to get a good impression of the targets. Faces were displayed until participants rated all characteristics and selected to move on to the next item. The order of emotion expressions was the same across all targets, and the order of targets was the same for each perceiver. 

Instead of asking participants to rate the 12 PCL:SV items, we instead asked them to only rate the items psychopathy, manipulativeness, and criminal behavior. We reduced the number of psychopathic characteristics to rate because, as was found in study 1, it was not clear that participants sufficiently differentiated between those traits in their ratings. We selected the item criminal behavior because it is a reflection of the social deviant factor, factor 2, of psychopathy. We selected the item manipulativeness because it represents the interpersonal-affective factor, factor 1, of psychopathy. We again included the items intelligence, attractiveness, trustworthiness, and masculinity. And we newly included sympathetic, responsibility, extraversion, and neuroticism. The items responsibility and sympathetic were included in addition to trustworthiness to further expand on ratings of traits that can be seen as opposites of psychopathy. The personality items, extraversion, and neuroticism were included because there is evidence that these traits can be perceived at zero acquaintance, and thus, they may also impact perceptions of psychopathy. 

Participants rated these 11 items using a 10-point Likert scale that ranged from 0 (very low) to 10 (very high). A broad description of every item was given to the participants before and during the task. Following the ratings, participants were asked to express how confident they were in the accuracy of their ratings; however, these results are excluded from the present study. 

**Identification of Emotion Expressions from Composite Faces (Emotion Composite).** A short version of the Emotion Composite Task (EmotCom) was administered, which consisted of 36 items (α = 0.46). These items were selected by choosing only two male and two female targets from the original version (normally, there are 8 targets).

**BEFKI gf.** The “Berliner Test zur Erfassung fluider und kristalliner Intelligenz” figural task ([38]) measures fluid intelligence. The task included 16 items in which the participants had to choose which pattern completes a series. The 16-trial task is scored as correct or incorrect, and the measure had acceptable internal consistency (α = 0.80).

### 5.2. Results

#### 5.2.1. Research Question 1: Can Psychopathy Be Perceived?

We estimated a social relations analysis to identify the variance in trait ratings attributable to the target and the responder. There was a significant proportion of variance attributable to the target for all traits, indicating there was something perceived in the target that predicted the responder’s ratings of the target (Table 3). 

The two traits associated with the individual psychopathy factors, namely manipulativeness and criminality, were above the 10% cut-off often used in the literature, ([26]). However, ratings of psychopathy were below 10%. As is typically found in the literature, there was a substantial portion attributable to the target for extraversion and attractiveness (e.g., [41]).

We again estimated accuracy in psychopathy perception with profile similarity analysis in which we compared the shape of ratings against the shape of the targets’ real scores ([34]; [33]). Ratings of psychopathy were correlated with the targets’ PCL:SV prorated total scores. Ratings of the item manipulativeness were correlated with the targets’ PCL:SV item-level scores for the item deceitful. Ratings of criminal behavior were correlated with the targets’ average PCL:SV item-level scores for adult antisocial behavior and adolescent antisocial behavior. Finally, ratings of responsibility were correlated with the targets’ PCL:SV item-level score for irresponsibility, which was reverse coded. 

On average, the correlations were positive but weak in magnitude, with person-level scores ranging from strongly negative to strongly positive (psychopathy: *M* = 0.11, ranged from −0.68 to 0.76; manipulativeness: *M* = 0.11, ranged from −0.58 to.76; criminal behavior: *M* = 0.05, ranged from −0.56 to 0.67; responsibility: *M* = 0.05, ranged from −0.61 to 0.63). 

#### 5.2.2. Research Question 2: Is the Perception of Psychopathy Due to Halo Effects?

Next, we re-estimated the social relations analysis for psychopathy, iteratively controlling for halo effects (see Table 4). Overall, we found that controlling for perceptions of other traits reduced the proportion of variance attributable to the target in ratings of psychopathic traits from 7% to 2% for psychopathy, from 14% to 5% for criminality, and from 20% to 3% for manipulativeness. In each case, the variance attributable to the target dropped to a non-significant value, indicating that the perception of psychopathic traits is related to the perception of other traits. 

#### 5.2.3. Research Question 3: Is Psychopathy Perception a Cognitive Ability?

To test whether accuracy in trait perception is related to cognitive abilities, like intelligence, we correlated accuracy scores for the perception of each trait. We estimated accuracy in the perception of intelligence with profile similarity analysis in which we compared the shape of intelligence ratings against the shape of the targets’ BEFKI figural task score real scores ([34]; [33]). On average, the correlations were low, again ranging from strongly negative to strongly positive for individual persons (intelligence: *M* = 0.06, ranged from −0.61 to 0.63). 

The accuracy in perception scores were correlated with each other and with the responder’s fluid intelligence and emotion perception ability (see Table 5). Based on the manifest correlations, we found that accuracy in the perception of psychopathy, manipulativeness, and responsibility were significantly correlated with fluid intelligence, while accuracy in the perception of criminality and intelligence were not. Emotion perception ability was unrelated to all of the trait perception accuracy scores. 

Because some effects may be attenuated by unreliability, we re-estimated these relations using latent variable models. Additionally, we evaluated measurement models of trait perception accuracy to test whether accuracy matches the dimensional structure found in general trait ratings. 

First, we modeled a single factor loading on accuracy in psychopathy perception, manipulativeness perception, criminality perception, intelligence perception, and responsibility perception (MM1). That model had a poor fit to the data (see Table 6). Next, we modeled two factors, one indicated by accuracy in the perception of socially desirable traits (intelligence and responsibility), and the other in socially undesirable traits (psychopathy, manipulativeness, and criminality) (MM2). That model had an excellent fit to the data (see Figure 3 for an illustration). We additionally tried two bifactor structures, creating a nested factor for either the socially desirable traits or the socially undesirable traits. Both had excellent fit to the data, but fewer degrees of freedom. Thus, we retained the correlated group factor model for our structural model.

In the structural model, we correlated both trait perception accuracy factors with each other and with fluid intelligence and emotion perception ability. The latter two constructs were modeled using parcels and a bifactor structure, with fluid intelligence as the reference factor and emotion perception as a nested factor. When examining correlations with emotion perception, the bifactor structure also allowed us to control for the influence of fluid intelligence. The model had excellent fit to the data (χ^2^_(16)_ = 14.4, *p* = 0.03, RMSEA = 0.105, CFI = 0.962, SRMR = 0.046). A bifactor structure is a commonly applied model for this kind of data (e.g., [21]; [51]), and it allows us to test incremental relations with emotion perception above that of fluid intelligence (see Figure 3). 

The structural model had excellent fit to the data (χ^2^_(37)_ = 43.90, *p* = 0.20, RMSEA = 0.038, CFI = 0.976, SRMR = 0.054), and all factors had acceptable factor saturation, as indicated by omega hierarchical estimates ([37]). Overall, we found both trait perception factors were positively correlated. Likewise, accuracy in the perception of socially undesirable traits was significantly correlated with general cognition but not emotion perception. Accuracy in the perception of socially desirable traits was unrelated to general cognition; however, it was significantly related to emotion perception ability (assuming a one-tailed *p* value). 

### 5.3. Discussion Study 2

Overall, we again found evidence that psychopathy could generally be perceived. There was a significant proportion of variance in ratings attributable to the target, and on average, participants’ ratings were weakly correlated to the targets’ PCL:SV scores. However, there was some variation for the individual factors. Namely, for criminal behavior and manipulativeness, there was a substantial portion of variance in the ratings attributable to the target. Because criminal behavior is part of factor 2 of the psychopathy definition, this result replicates what we found in study 1. In contrast, we found a stronger proportion of variance attributable to the target for manipulativeness, which is part of factor 1, relative to what was observed in study 1. The stronger proportion of variance could be due to the definitions that we gave the responders that explicitly described the items. In study 1, we used item descriptions similar to the ones of the PCL, whereas, in study 2, we tried to use more common expressions to describe the items, which could have helped the responders to better understand what trait they were rating. Using the item “psychopathy” could still have been difficult to rate for the responders because this term is stigmatized, which would explain why it was still beyond the 10% cut-off.

Like in study 1, we found halo effects in our ratings, which means the effect of significant variance components for the traits of psychopathy, criminal behavior, and manipulativeness nearly disappears after controlling for the other traits. 

Then, we checked whether trait perception is a cognitive ability and found that fluid intelligence (gf) was significantly positively correlated with the ability to perceive psychopathy in general, manipulativeness, and responsibility, as well as emotion perception. Emotion perception was not significantly correlated with the ability to perceive any of the traits but correlated with the ability to perceive positive traits at trend level. General cognition, on the other hand, was significantly correlated with the ability to perceive negative traits. Because manifest correlations are attenuated by reliability, we next estimated a structural equation model relating emotion perception, fluid intelligence, and the perception of socially desirable and socially undesirable traits. We found that the trait perception factors for socially desirable and socially undesirable traits were positively correlated. The model also showed that the accuracy in the perception of socially desirable traits was significantly correlated with general cognition but not emotion perception, which was correlated with the accuracy in the perception of socially undesirable traits. The accuracy in the perception of socially undesirable traits was however unrelated to general cognition. 

Overall, study 2 largely replicated the findings of study 1, showing that psychopathy could be perceived by responders. As in study 1, we found halo effects for the individual traits, but some of them still remain significant, even after controlling for halo effects. In study 2, it was possible to explore the connection between other cognitive abilities and person perception and to answer the question of whether it is a cognitive ability. Although the perception of undesirable traits is not correlated with emotion perception, there is a significant relationship with general cognition. Furthermore, the perception of socially desirable traits is correlated with emotion perception close to significance. These results support the hypothesis of psychopathy perception being a cognitive ability.

## 6. General Discussion

We investigated whether individuals could perceive psychopathy and whether the ability to perceive psychopathy is a cognitive ability. Previous studies exploring the detection of psychopathy were partly limited to non-offender samples, whereas we used prison inmates and criminal offenders in other institutions as targets. One study showed that the shorter the time periods in which psychopath stimuli were presented, the more accurate were the ratings ([14]), which supports our study design of only using static pictures. The current research advanced these studies by developing a novel task on which responders rated targets with zero acquaintance on several different attributes, including psychopathy, attractiveness, and intelligence. 

Overall, while data collection occurred in two different countries (i.e., the United States of America and Australia) and with different sample characteristics (i.e., an online adult sample and a university student sample), we nevertheless found replication of effects. In the following sections, we will discuss the results organized by research question. 

### 6.1. Research Question 1: Can Psychopathy Be Perceived?

In both studies, we found significant agreement between the responders for all aspects of psychopathy. We also found that perceivers were, on average, accurate in their perception of psychopathy. They were also accurate in their perception of intelligence, as indicated by weakly positive correlations between trait ratings and veridical measures of those traits. [23] ([23]) and others have speculated as to the cause of this relation. First, it is possible that the personality trait of psychopathy is dually inherited with specific physical features. Alternatively, it may be that individuals with specific facial features are perceived by themselves or others to have a specific personality, which influences their personality development. 

Despite the evidence provided here and elsewhere that individuals are more accurate than chance in their capacity to identify a highly psychopathic person, we would like to strongly caution against a Lombrosian perspective on these results. There were strong differences between persons in their capacity to identify psychopathy. Many individuals not only had no relation between their psychopathy ratings and the targets’ PCL:SV scores, which would indicate trait ratings are made randomly, but also several had *negative* correlations, indicating that the lower men were on psychopathy, the higher these individuals rated them on psychopathy. Thus, one should generally be very skeptical of any "gut" feelings about another person’s psychopathy levels, based on a thin slice or single picture of that person. 

Additionally, we and others have not provided any clear evidence as to which facial features indicate psychopathy. It could be certain craniofeatures, as [23] ([23]) and others have suggested. It may also be that their neutral expression has traces of prototypical anger expressions, which are interpreted as psychopathy. 

### 6.2. Research Question 2: Is the Perception of Psychopathy Due to Halo Effects? 

In both studies, we found that halo effects heavily influence the perception of psychopathy, which means the perception of psychopathy is related to how the participants perceived other traits. Specifically, perceptions of a targets’ attractiveness, masculinity, intelligence, and extraversion impacted perception. In general, this is to be expected, as nearly all personality traits have halo effects (e.g., [41]), and others have found that individuals perceived to be highly psychopathic are also perceived to be less attractive ([46]; [3]). Future research should expand on this point and investigate how these other features impact psychopathy ratings. 

### 6.3. Research Question 3: Is Psychopathy Perception a Cognitive Ability?

In both studies, we found generally positive correlations between trait perception abilities, with some distinction between the ability to undesirable and desirable traits. In both studies, there was a significant positive correlation between accuracy in the perception of undesirable traits: Study 1, between psychopathy factor 1 and psychopathy factor 2; and Study 2, between psychopathy, manipulativeness, and criminality. In both studies, there was also no correlation between the accuracy in the perception of undesirable and desirable traits. In study 1, this was shown through a zero correlation between accuracy in the perception of the psychopathy factors and accuracy in the perception of intelligence. In study 2, this was replicated through generally non-significant correlations between accuracy in the perception of intelligence and responsibility with accuracy in the perception of psychopathy, manipulativeness, and criminality. Through structural equations modeling, we found accuracy in trait ratings could be organized within two factors, which we labeled as accuracy in the perception of undesirable or desirable traits. 

Accuracy in the perception of undesirable traits was composed of psychopathy characteristics and was found to be moderately related to higher general cognition, suggesting it may be a cognitive ability. Accuracy in the perception of desirable traits was then found to be positively correlated with emotion perception ability (and significantly associated, assuming a one-tailed *p* value), which replicates the findings of Study 2. 

While there are several possible reasons for this pattern of relations, we will offer a tentative suggestion. Emotion perception is identified as an integral component in emotional intelligence ability ([36]), a cognitive ability that supports interpersonal relationships. It may be that individuals with these socio-emotional skills are more likely to focus on the positive traits of others and thus develop a stronger ability to accurately rate those traits. 

In the perception of undesirable traits, or psychopathy more specifically, there may be a different mechanism at work. As evolutionary psychologists have discussed, the capacity to identify a psychopath in one’s group is extremely important for detecting cheating and for the group’s success. However, the low prevalence rate of psychopathy, 1% of the male population, is often argued as one reason why psychopathy has persisted; namely, the rate is low enough to avoid general detection. Because of the importance of identifying highly psychopathic individuals, it may be that individuals who have stronger general cognitive abilities develop this capacity. 

### 6.4. Limitations and Future Directions

The two studies have various strengths, especially in the methodology of the person-perception task, as it is a newly created task with pictured from highly psychopathic offenders. However, as we explored what our responders agreed on in the ratings and whether perception errors occurred in the ratings, in this study, we did not explore the question of why the responders rated the way they did. For future studies, this might be an interesting topic to investigate. Furthermore, the person–perception task could be improved by adding female faces or by analyzing the faces beforehand to identify which faces have common characteristics and then to check whether they are rated similarly. 

The present study was limited to the perception of psychopathy in white male faces. Because male faces are generally perceived as more threatening relative to female faces ([20]) and because black faces can be perceived as more threatening relative to white faces (e.g., [24]), these results cannot be generalized to the perception of psychopathy in either group.

### 6.5. Conclusions

Overall, while we found support for a general ability to perceive psychopathy from the faces of others, the strong individual differences in accuracy suggest one should generally be very skeptical of any "gut" feelings about another person’s psychopathy levels based on a thin slice or single picture of that person. We also found some evidence that accuracy in psychopathy perception is related to general cognition, suggesting it may be a cognitive ability. However, more research is needed in this area. Specifically, it is important for this field to explore the facial cues that might have influenced the ratings and, in turn, to dissect more specifically which facial or behavioral cues might indicate danger. 

## Figures and Tables

**Figure 1 jintelligence-09-00029-f001:**
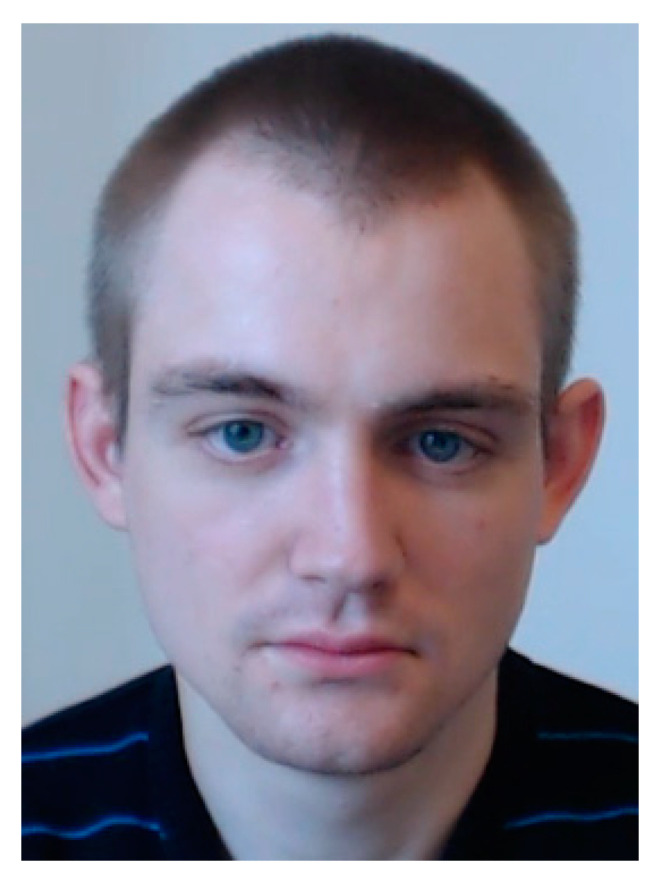
Example of the person perception task.

**Figure 2 jintelligence-09-00029-f002:**
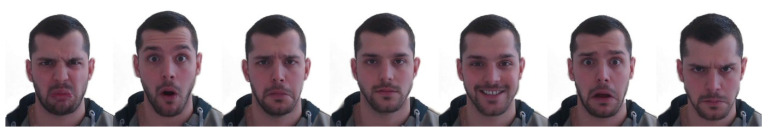
Target showing the emotion expressions in the Person-Perception Task.

**Figure 3 jintelligence-09-00029-f003:**
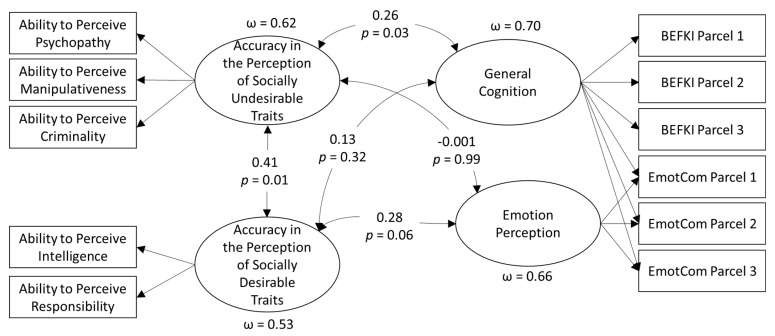
Structural Model Relating Person Perception accuracy with Cognitive Abilities.

**Table 1 jintelligence-09-00029-t001:** Study 1: Social Relations Analysis with and without controlling for halo effects.

Trait	Variance		Percent of the Total Variance	
	Responder	Target	Responder	Target
Attractiveness	0.81 *	0.42 *	31%	16%
Masculinity	0.83 *	0.13 *	33%	5%
Trustworthiness	0.51 *	0.19 *	20%	8%
Intelligence	0.54 *	0.21 *	23%	9%
Psychopathy Factor 1	0.59 *	0.13 *	36%	8%
Controlling for attractiveness	0.57 *	0.13 *	35%	8%
Controlling for attractiveness and masculinity	0.55 *	0.11 *	35%	7%
Controlling for attractiveness, masculinity, and intelligence	0.55 *	0.11 *	35%	7%
Controlling for attractiveness, masculinity, intelligence, and trustworthiness	0.55 *	0.07 *	35%	5%
Psychopathy Factor 2	0.70 *	0.29 *	33%	14%
Controlling for attractiveness	0.75 *	024 *	37%	12%
Controlling for attractiveness and masculinity	0.74 *	0.22 *	37%	11%
Controlling for attractiveness, masculinity, and intelligence	0.74 *	0.17 *	37%	9%
Controlling for attractiveness, masculinity, intelligence, and trustworthiness	0.76 *	0.12 *	42%	7%

Note: *n* = 170[note 1], * *p* < 0.05.

**Table 2 jintelligence-09-00029-t002:** Study 1: Manifest correlations of trait perception abilities with emotion perception ability.

	Accuracy in Perceiving Psychopathy Factor 1	Accuracy in Perceiving Psychopathy Factor 2	Accuracy in Perceiving Intelligence
Emotion perception ability	0.19 *p* = 0.07	0.10 *p* = 0.35	**0.23** ***p* = 0.03**
Ability to perceive intelligence	0.01 *p* = 0.93	0.00 *p* = 0.97	
Ability to perceive psychopathy Factor 2	**0.23** ***p* = 0.03**		

Note: *n* = 89, statistically significant coefficients (two-tailed *p* < 0.05) are presented in boldface.

**Table 3 jintelligence-09-00029-t003:** Study 2: Social Relations Analysis of trait ratings.

Trait	Variance		Percent of the Total Variance	
	Responder	Target	Responder	Target
Extraversion	0.87 *	0.98 *	17%	19%
Sympathetic	1.41 *	0.31 *	30%	7%
Trustworthy	1.32 *	0.39 *	30%	9%
Intelligence	1.34 *	0.28 *	30%	6%
Attractiveness	1.92 *	0.96 *	35%	17%
Masculinity	1.58 *	0.97 *	28%	17%
Neuroticism	1.45 *	0.20 *	31%	4%
Responsible	1.38 *	0.31 *	34%	8%
Manipulativeness	1.19 *	0.39 *	60%	20%
Criminal behavior	1.67 *	0.91 *	26%	14%
Psychopathy	1.67 *	0.38 *	33%	7%

Note: *n* = 126, * *p* < 0.05.

**Table 4 jintelligence-09-00029-t004:** Study 2: Social Relations Analysis of psychopathic traits after iteratively controlling for halo effects: Percent of variance attributable to the target.

	Perception of Psychopathy	Perception of Criminal Behavior	Perception of Manipulativeness
Original	7% *	14% *	20% *
Controlling for attractiveness	8% *	14% *	8% *
Controlling for attractiveness and masculinity	8% *	13% *	8% *
Controlling for attractiveness, masculinity, and intelligence	7% *	11% *	7% *
Controlling for attractiveness, masculinity, intelligence, and extraversion	7% *	10% *	8% *
Controlling for attractiveness, masculinity, intelligence, extraversion, and neuroticism	3%	8% *	6% *
Controlling for attractiveness, masculinity, intelligence, extraversion, neuroticism, and sympathetic	2%	6% *	4% *
Controlling for attractiveness, masculinity, intelligence, extraversion, neuroticism, sympathetic, and trustworthiness	2%	5% *	3% *

Note: *n* = 126, * *p* < 0.05.

**Table 5 jintelligence-09-00029-t005:** Study 2: Manifest Correlations between cognitive abilities and trait perception abilities.

	Emotion Perception Ability	Accuracy in Psychopathy Perception	Accuracy in Manipulativeness Perception	Accuracy in Criminality Perception	Accuracy in Intelligence Perception	Accuracy in Responsibility Perception
Fluid Intelligence	**20** ***p* = 0.03**	**0.20** ***p* = 0.03**	**0.27** ***p* = 0.004**	0.13 *p* = 0.16	−0.06 *p* = 0.53	**0.20** ***p* = 0.03**
Emotion Perception Ability		−0.08 *p* = 0.39	0.13 *p* = 0.15	−0.02 *p* = 0.81	0.13 *p* = 0.16	0.15 *p* = 0.10
Accuracy in Psychopathy Perception			**0.32** ***p* < 0.001**	**0.32** ***p* < 0.001**	0.11 *p* = 0.22	0.07 *p* = 0.46
Accuracy in Manipulativeness Perception				**0.40** ***p* < 0.001**	0.12 *p* = 0.20	0.13 *p* = 0.15
Accuracy in Criminality Perception					**0.22** ***p* = 0.01**	**0.20** ***p* = 0.03**
Accuracy in Intelligence Perception						**0.36** ***p* < 0.001**

Note: *n* = 126.

**Table 6 jintelligence-09-00029-t006:** Study 2: Model fit of the trait perception ability measurement models.

Model	χ^2^	df	CFI	RMSEA	AIC
MM1) Single general trait perception factor	13.5 *	5	0.851	0.118	183
MM2) Correlated socially desirable and undesirable trait perception factors	1.5	5	1.000	0.000	172
MM3) Bifactor model with nested desirable trait factor	1.5	4	1.000	0.000	173
MM4) Bifactor model with nested undesirable trait factor	0.3	2	1.000	0.000	176

Note: * *p* < 0.05.

## Data Availability

The data is available from the authors by request.

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
