# Peer review of "Detection of Psychopathic Traits in Emotional Faces"

_jintelligence, 2021, doi:10.3390/jintelligence9020029_

Round 1

Reviewer 1 Report

This paper describes two studies examining if high psychopathy traits are perceivable and whether other traits affect ratings of psychopathic traits, like a halo effect. They investigate the impact of attractiveness, masculinity, intelligence, trustworthiness, extraversion, sympathy and responsibility in causing halo effects in perception of psychopathy. They also consider if personality traits and cognitive abilities (fluid intelligence and emotion perception ability) of the perceiver affect accuracy in perceiving psychopathic traits?

To test their hypothesis, they have participants rate psychopathic traits in various faces of men. For the perception task in Study 1, each face is rated (Likert scale 1-7) on 16 traits, including attractiveness, intelligence, masculinity, trustworthiness. Importantly, definitions of these terms are provided before the test. From these ratings determine scores in Factor 1: psychopathy superordinate factor 1 (superficial, grandiose, deceitful, lacks remorse, lacks empathy, doesn’t accept responsibility) factor 2 (impulsive, lacks goals, irresponsible, poor behavior controls, adolescent antisocial behavior, adult antisocial behavior). They also rate participant’s own psychopathy levels, careful to use a standard assessment which does not suffer from personal biases in self-reports or biased reports from others. They evaluate agreement between raters, proportion of variance in ratings based on perceiver’s own psychopathy, the influence of halo effects, and whether accuracy is related to cognitive abilities, using a profile similarity metrics to examine correlations between ratings of psychopathy and viewer’s own psychopathy level.

In Study 1, they found large individual differences in perceiving psychopathy and the presence of halo effects, suggesting that ratings of psychopathic traits were not independent and were influenced by other dimensions of the face. The findings for correlations between cognitive ability and psychopathy ratings were weak and  mixed. Given the variability of results from Study 1 and confounding variables such as distracting items on the face, such as glasses, or information in the background, they replicated the experiment in Study 2, recruiting a larger sample with a narrower age range and using a better sample of faces to present a subset of stimuli from the original sample. They also had participants rate fewer dimensions for each face, instead of the original 16 traits, they only assessed 11 traits on a 1-10 point Likert scale, adding some new dimensions, such as extroversion and neuroticism. It is unclear if Study 2 was also online or conducted in a lab setting as line 296 says the next study was conducted in a lab but line 315 says Study 2 was administered online.

In Study 2, the authors found that accuracy in the perception of psychopathy, manipulativeness, and responsibility correlated with fluid intelligence but accuracy in perception of criminality and intelligence did not correlate and emotion perception ability did not relate with any measure of trait perception. Overall, psychopathy could not be perceived.

Major issues:

  1. The authors present important limitations of previous work in the introduction. The measures and assessments are fairly clear, although methods for the behavioral task need more detail. How long was a given face displayed? How much time did participants have to reply? Was order of face presentation randomized? Was the order of perception test and other assessments randomized? Some of these procedures are important as the authors claim that average accuracy was quite low on line 290 and it is not clear why.
  2. Study 1 was conducted in 331 participants, of which only 89 rated all faces and 170 rated less than 10 faces. Yet, analyses were sometimes, but not always, done on all 170 participants, in which case different numbers of trials are being contributed for each face due to variation in the number of participants. The authors claim they can still quantify medium effects sizes with the smaller sample, but it is not clear how the results would vary.  Of note, the authors state that they quantified correlations between psychopathy perception and cognitive ability only in the subset of 89 participants who had rated all faces, to ensure stable estimates. Why was this not important for the earlier analysis in Study 1? Furthermore, on line 286, the results for correlations between accuracy in the perception of psychopathy factor 1 and accuracy in the perception of emotion report a one-tailed test, but results should reflect a two-tailed p value, as there is no strong rationale for using a one-tailed test in this one case. Importantly, for Study 2, a larger sample of 126 participants completed all aspects of the experiment. Study 2 replicated Study 1, for the most part, finding halo effects and weak correlations between perception of psychopathic traits and emotion perception.
  3. Differences between Study 1 and Study 2 need to made clearer in the abstract and main text and there should be some discussion of the influence of culture on these effects as Study 1 and 2 used students from different cultures. It would be clearer for the reader if Study 1 was included as a pilot study with 89 participants and Study 2 was the main study with 126 participants. There are also many changes in faces presented and traits measured without enough justification. Finally, if Study 2 did not group traits in factor 1 and factor 2, did it have enough power given the sample size to consider the effects of interest? Why did the authors do away with the breakdown into factor 1 and factor 2?
  4. The authors need to state their findings more clearly. the Abstract ends on line 17: ”Accuracy in perception was weakly or unrelated with fluid intelligence and emotion perception ability”. The word "respectively" needs to be added to end this sentence to agree with the Conclusion on page 16, line 143: “Accuracy in the perception of socially undesirable traits was positively correlated with fluid intelligence but not with emotion perception.”

Minor issues:

Grammatical errors and/or typos are outlined below:

The authors should be consistent in their use of terminology: always use the same term, “thin-slice” with a hyphen or never use the hyphen.

Line 41: should this read “thin slices” instead of “thin slides”?

Line 86: “….ratings of the targets, and that that some targets are…”

Line 144: “that stems from a diverse ample …”

Reviewer 2 Report

The construct of psychopathy has been discussed for quite a long time and the authors solely remain on the first model of psychopathy presented By Robert Hare in the early nineties. It would be important to acknowledge the relevance of other theoretical models, even from Hare, but mainly from David Cooke and associates that portray psychopathy as a three dimensional model.

In addition, it would be important that the authors made a clear statement that their research is far from a Lombrosian perspective and state clearly the implications that their findings could have for criminal policy and criminal research.

Reviewer 3 Report

In this article the author(s) are interested in assessing the ability of people to perceive the psychopathy in others they have not met by only viewing their face. This is quite an interesting question and I found it fascinating, but I have several significant concerns about the manuscript.

My first major concern relates to the ability of people to perceive psychological disorders like psychopathy. I am not sure of that I even believe this is a realistic possibility. The only way this seems like it could be possible is through exterior features that provide cues to perceivers. Perhaps some set of exterior features such deep set eyes, oval shaped heads, etc. are more common among people who have psychopathy and we have been trained by evolution to recognize these cues, but this seems a stretch. Additionally, there was no discussion of how people might make such judgements about people. Is it just their gut? There was a discussion that previous research showed people could make such judgements with some level of accuracy, but some empirically driven discussion of how people might perceive such psychological features seems warranted.

The author(s) note in their limitations that female faces were not included and should be in future research. I would also suggest Black faces or faces of color should be included too and that they are not in this research is a major concern. How do the author(s) account for the influence of race and gender? Previous research has shown that men are perceived as more threatening, Black men are perceived as even more threatening than their White counterparts. This relates back to my concern about how these perceptions are made by the perceivers. At the very least the author(s) should devote some space to a discussion of how the results were influenced by the use of white, male faces and how it limits their findings.

The author(s) don’t give much attention to what I think is their most interesting finding and relates to my concern of how people are building their perceptions, the halo effect. I think the author(s) could spend a lot more space discussing the importance of these factors (i.e. attractiveness) that inform how we build our perceptions of people.

Finally, the author(s) spend almost the entirety of their discussion sections simply restating the findings they noted in their findings sections. I would have liked to see a discussion of what the results tell us generally. How are they informative? How do they help us understand how people build their perceptions of people? What explanation is there of why these findings are present? By my assessment the authors(s) give one sentence of such an explanation in the entire manuscript. The author(s) simply do not answer the “so what” question. So, people can perceive psychopathy (somewhat) by looking at faces – why is this valuable, how do they do it, and so what?

Reviewer 4 Report

I wrote some commentaries in the MS. 

Without a deep view of the measure that they used, it is hard to understand some points that they argued in the MS. 

My main concern is regarding the IQ measure of the test.. It is necessary more evidence of how they have determined that accuracy is part of CHC (Cattell Horn Carroll Cognitive Mental Abilities). It is true that perceiving is part of the first stratum of CHC IQ postulates, but it is not part of the Gf (fluid intelligence), which is at the second stratum.  Can they provide some evidence of concurrent validity of the face test to IQ measures?

The rest of the commentaries were set in the MS-pdf file.

I liked the paper, it is pretty exciting and challenging.

Round 2

Reviewer 1 Report

The manuscript is improved and the authors have addressed my major and minor concerns.

Reviewer 2 Report

The paper has been improved and addressed my previous major concerns, namely in the partial conclusions and in the final discussion.

Reviewer 3 Report

Overall the manuscript is improved and I suggest it be accepted in its present form. 

Reviewer 4 Report

accept in the present form